# Roles of the membrane-reentrant β-hairpin-like loop of RseP protease in selective substrate cleavage

**Koichiro Akiyama[1†], Shinya Mizuno[1†], Yohei Hizukuri[1], Hiroyuki Mori[1], Terukazu Nogi[2], Yoshinori Akiyama[1]***

[1]Institute for Virus Research, Kyoto University, Kyoto, Japan; [2]Graduate School of Medical Life Science, Yokohama City University, Yokohama, Japan

**Abstract** Molecular mechanisms underlying substrate recognition and cleavage by *Escherichia coli* RseP, which belongs to S2P family of intramembrane-cleaving proteases, remain unclear. We examined the function of a conserved region looped into the membrane domain of RseP to form a β-hairpin-like structure near its active site in substrate recognition and cleavage. We observed that mutations disturbing the possible β-strand conformation of the loop impaired RseP proteolytic activity and that some of these mutations resulted in the differential cleavage of different substrates. Co-immunoprecipitation and crosslinking experiments suggest that the loop directly interacts with the transmembrane segments of substrates. Helix-destabilising mutations in the transmembrane segments of substrates suppressed the effect of loop mutations in an allele-specific manner. These results suggest that the loop promotes substrate cleavage by selectively recognising the transmembrane segments of substrates in an extended conformation and by presenting them to the proteolytic active site, which contributes to substrate discrimination.

*For correspondence:
yakiyama@virus.kyoto-u.ac.jp

†These authors contributed equally to this work

## Introduction

Regulated intramembrane proteolysis (RIP) is a common mechanism for transmembrane signalling and is a part of many important cellular processes in both prokaryotes and eukaryotes (*Ha, 2009*; *Wolfe, 2009*; *Urban, 2013*). Proteases involved in RIP are called intramembrane-cleaving proteases (I-CLiPs) and catalyse the proteolytic cleavage of transmembrane segments (TMs) of target membrane proteins within the lipid bilayer. *Escherichia coli* RseP belongs to S2P zinc metalloproteinase family of I-CLiPs. It plays a key role in regulating the σ^E extracytoplasmic stress response (*Hizukuri et al., 2013*; *Kroos and Akiyama, 2013*), in which the dedicated sigma factor σ^E is activated through the successive cleavage of a single membrane-spanning anti-σ^E RseA in response to cell surface stresses. The first cleavage catalysed by DegS (site-1 cleavage) on the periplasmic side triggers the intramembrane cleavage by RseP (site-2 cleavage) (*Alba et al., 2002*; *Kanehara et al., 2002*). Recent studies suggest that in addition to being involved in RseA cleavage, RseP acts in the proteolytic removal of remnant signal peptides from the membrane (*Saito et al., 2011*). RseP spans the membrane 4 times, with both the termini facing the periplasm. Its central periplasmic region contains tandem PDZ domains (PDZ-N and PDZ-C; *Figure 1A*) (*Kanehara et al., 2001*; *Kinch et al., 2006*; *Inaba et al., 2008*). The S2P family proteases share a conserved core domain containing 3 TMs. In RseP, these correspond to TM1–TM3 (*Figure 1A,B*) (*Kinch et al., 2006*). The first and the third TM contain HExxH and LDG active site motifs, respectively. Disulfide crosslinking and co-immunoprecipitation experiments suggest that TM3 plays a critical role in binding of a substrate TM (*Koide et al., 2008*). Helix destabilisation of substrate TMs promotes their binding to and cleavage by RseP (*Akiyama et al., 2004*; *Koide et al., 2008*).

**eLife digest** Cells have communication systems that enable them to respond to potentially dangerous changes in their external environment. For example, bacteria have an enzyme called RseP that helps to activate responses to external stresses. This enzyme sits in the membrane that surrounds the cell and cuts a protein called RseA to release a signal molecule into the cell interior. This signal molecule then promotes the expression of particular genes to protect the cell from harm.

Previous studies have identified the 'active site' of RseP, which is the region of the enzyme that actually cuts the target protein. However, it is not clear how the enzyme is able to identify and cleave RseA and its other 'substrate' proteins. The enzyme also contains a structure called a β-hairpin-like loop that is close to the active site, which is not commonly found in membrane proteins. Here, Akiyama, Mizuno et al. used genetic and biochemical techniques to study the role of this loop structure in the RseP enzyme of the *E. coli* bacterium.

The experiments show that the loop specifically binds to a section of substrate proteins—called the transmembrane segment—that spans the cell membrane. Several genetic mutations that affected the loop altered the ability of RseP to bind to and cleave substrates. The effect of these mutations in RseP could be suppressed by introducing genetic mutations in substrates that altered the transmembrane segments. Akiyama, Mizuno et al. propose that the β-hairpin-like loop of the RseP enzyme binds the transmembrane segment of a substrate and presents it to the active site.

A previous study showed that another region of RseP called the periplasmic PDZ domains can act as a filter to stop RseP cutting other membrane proteins in error. Akiyama, Mizuno et al.'s findings suggest that the β-hairpin-like loop serves as an additional checkpoint to identify RseA and other proteins that RseP targets. The next step is to carry out further experiments to test this model.

RseP-catalysed site-2 cleavage of RseA depends on site-1 cleavage (*Alba et al., 2002*; *Kanehara et al., 2002*), which results in stress-dependent σ$^E$ activation, because site-1 cleavage is induced by stress signals (*Walsh et al., 2003*; *Kulp and Kuehn, 2011*; *Lima et al., 2013*). Previous studies suggest that RseP PDZ domains are involved in the site-1 cleavage-dependence of the site-2 cleavage by RseP (*Kanehara et al., 2003*; *Bohn et al., 2004*; *Grigorova et al., 2004*; *Inaba et al., 2008*; *Hizukuri and Akiyama, 2012*). Although the crystal structure of an S2P homolog (mjS2P) from *Methanocaldococcus jannaschii* (*Feng et al., 2007*) has been reported, it does not have a PDZ domain, and the three-dimensional structure of an S2P homolog with PDZ domain(s) is not available. We recently reported the crystal structure of the PDZ tandem of an *Aquifex aeolicus* RseP homolog. The structure suggests that the 2 PDZ domains create a single pocket-like structure that covers the core membrane domain on the membrane surface (*Hizukuri et al., 2014*). Structural models and biochemical and genetic results suggest that the PDZ tandem acts as a size-exclusion filter to allow only the periplasmically truncated form of RseA to enter the recessed active site in the membrane domain of RseP (*Kroos and Akiyama, 2013*; *Hizukuri et al., 2014*). However, mechanisms underlying substrate proteolysis by RseP remains elusive. It is unclear whether the size-exclusion function of the PDZ tandem is sufficient to discriminate between substrates and non-substrates and how a substrate TM, which assumes an α-helical conformation that needs to be unwound for proteolysis, is recognised and presented to the proteolytic active site. The mechanism of substrate discrimination and specific cleavage in RIP is one of the major problems to be solved (*Langosch et al., 2015*). Detailed knowledge of mechanisms underlying substrate discrimination and cleavage would help in controlling the cleavage of membrane proteins by I-CLiPs including S2P proteases.

The structure of mjS2P indicates that it contains a region, located close to the active site, that is looped into the membrane domain from the cytoplasmic side (*Figure 1B,C*). This membrane-reentrant loop is unique because it is in an extended or β-strand conformation, whereas a membrane-embedded polypeptide is generally in an α-helical conformation. Therefore, we have named this region membrane-reentrant β-loop (MRE β-loop). The MRE β-loop is conserved in proteins belonging to groups I and III of S2P subfamilies, including RseP (group I), *Bacillus subtilis* SpoIVFB (group III) and mjS2P (group III; *Figure 1D*) (*Ha, 2009*; *Kroos and Akiyama, 2013*; *Zhang et al., 2013*). A recent study showed that the MRE β-loop of SpoIVFB can be crosslinked with a substrate through disulfide bonds, suggesting its possible involvement in substrate interaction (*Zhang et al., 2013*). However,

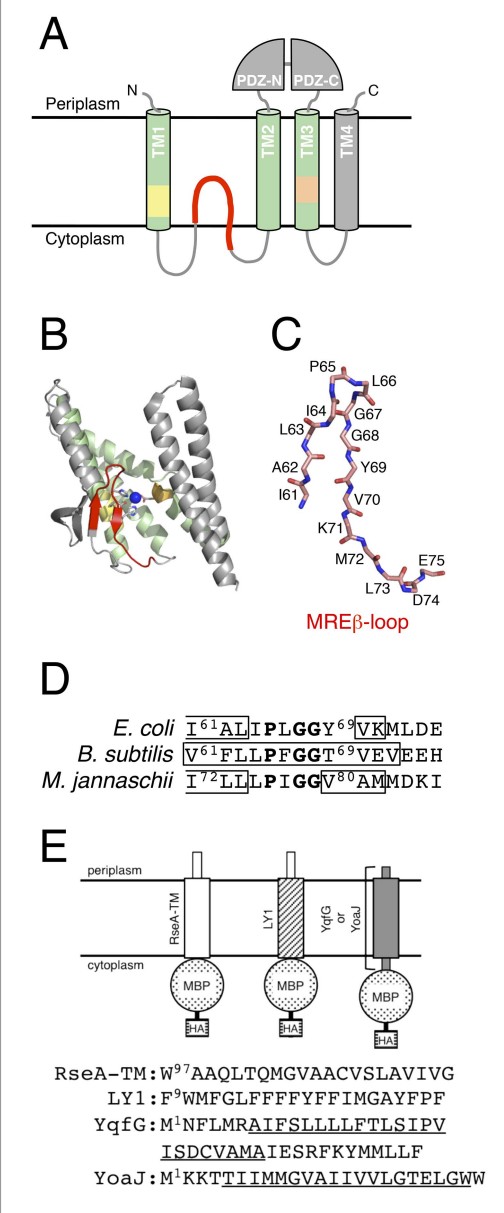

**Figure 1**. Structures of RseP and the model substrates used in this study. (**A**) Schematic representation of RseP. (**B**) Crystal structure of mjS2P (PDB code: 3B4R A). The secondary structure was analysed using STRIDE (**Frishman and Argos, 1995**), and the image was generated using PyMOL (https://www.pymol.org) (**Schrödinger, 2010**). In **A** and **B**, the 3 TM helices composing the core membrane domain are shown in light green; the MRE β-loop, HExxH motif and LDG motif are shown in red, yellow and orange, respectively, and the coordinated zinc atom is shown in blue. (**C**) The structure of the mjS2P MRE β-loop. The polypeptide backbone is extracted from the model in **Figure 1B** and shown as sticks. C, O and N atoms are shown in pink, red and blue, respectively. The amino acid residues of the corresponding region of RseP are assigned on the model. (**D**) Sequences of the MRE β-loop of RseP

*Figure 1. continued on next page*

functional roles of the MRE β-loop remain largely unknown. Here, we analysed the function of the RseP MRE β-loop and found that it could directly bind to and promotes selective cleavage of RseP substrates. Therefore, we propose that the MRE β-loop stabilises substrate TMs in an extended conformation and presents them to the proteolytic active site.

## Results

### Importance of the MRE β-loop in the proteolytic function of RseP

We performed systematic mutational analysis to investigate the possible role of the MRE β-loop in RseP function. We constructed an MRE β-loop deletion mutant (Δloop mutant) containing a short linker (Gly–Gly) in place of residues Ile-61 to Glu-75. Accumulation level of the Δloop mutant of RseP-HM (RseP with a C-terminal His-Myc bipartite tag) was comparable to that of wild-type RseP-HM (**Figure 2**), suggesting that the Δloop mutation did not cause global structural changes in RseP. The Δloop mutant did not exhibit complementation activity against a Δ*rseP* mutation (**Figure 2A**). Its protease activity was examined using model substrates HA-MBP-RseA148 and HA-MBP-RseA(LY1)148 (**Figure 2C,D**). HA-MBP-RseA148 is a derivative of the DegS-cleaved form of RseA (RseA148) and contains a maltose-binding protein (MBP) domain with an N-terminal haemagglutinin (HA) tag in place of the cytoplasmic region of RseA (**Figure 1E**) (**Hizukuri and Akiyama, 2012**). HA-MBP-RseA (LY1)148 is essentially similar to HA-MBP-RseA148, except that it has the first TM (LY1) of lactose permease (LacY) instead of the RseA TM (**Figure 1E**) (**Hizukuri and Akiyama, 2012**). Our previous studies showed that an N-terminally-attached tag has little effect on substrate cleavage by RseP (**Akiyama et al., 2004**; **Saito et al., 2011**). The model substrates were co-expressed with RseP-HM or its derivatives in a Δ*rseA*/Δ*rseP* strain. The substrates were converted from the full-length form (FL) to a cleaved form (CL) after co-expression with wild-type RseP but not with its proteolytically inactive mutant having an amino acid alteration (E23Q) in the H$^{22}$ExxH active site motif, indicating that these substrates underwent RseP-mediated proteolysis, as described previously (**Hizukuri and Akiyama, 2012**). Co-expression of the substrates with the Δloop mutant did not convert them from FL to CL, indicating that the mutant was proteolytically inactive.

*Figure 1. Continued*

(*E. coli*), SpoIVFB (*B. subtilis*) and mjS2P (*M. jannaschii*). The regions predicted to form β-strands are boxed. The secondary structure is assigned based on the analysis of amino acid sequences by PsiPred (http://bioinf.cs.ucl.ac.uk/psipred) (for RseP and SpoIVFB) or the analysis of the crystal structure by STRIDE (for mjS2P). The sequence alignment and secondary structure prediction suggest that this region of RseP also assumes the β-hairpin-like structure. The conserved PxGG motif is boldfaced. (**E**) Schematic representation of the model substrates and their amino acid sequences. RseA- and LacY-TM1 (LY1)-derived regions are shown as white and hatched rectangles, respectively. YqfG- or YoaJ-derived region is shown as a gray rectangle. Amino acid sequences of RseA and LY1 TMs and the entire region of YqfG and YoaJ are shown. The possible TMs of YqfG and YoaJ are underlined.

We next examined the effects of MRE β-loop mutations on the cleavage of other substrates. Recent studies have shown that some small membrane proteins are encoded by the *E. coli* genome and expressed as functional proteins (*Hemm et al., 2008*; *Alix and Blanc-Potard, 2009*; *Fontaine et al., 2011*). YqfG and YoaJ, small membrane proteins with unknown functions, are predicted to have type II ($N_{IN}$-$C_{OUT}$) membrane orientation according to the TMHMM program (*Krogh et al., 2001*). Membrane topology of YoaJ was confirmed experimentally (*Fontaine et al., 2011*). We examined the cleavage of these proteins by RseP because their structural features were analogous to those of signal peptides that were recently identified as RseP substrates (*Saito et al., 2011*).

To facilitate detection, we attached an HA-MBP domain to the N-termini of YqfG and YoaJ (*Figure 1E*) and expressed them in Δ*rseA*/Δ*rseP* strain (*Figure 2E,F*). Anti-HA immunoblotting showed the accumulation of a protein of the expected size (approximately 46 kDa) when HA-MBP-YqfG was expressed alone (*Figure 2E*). The protein was converted to a slightly smaller fragment upon co-expression with wild-type RseP-HM but not with RseP-HM E23Q mutant, indicating that HA-MBP-YqfG was cleaved by RseP presumably within its TM. As expected, the Δloop mutant did not cleave HA-MBP-YqfG. In contrast, HA-MBP-YoaJ did not undergo any cleavage (*Figure 2F*). Cell fractionation and alkali extraction showed that most HA-MBP-YoaJ was integrally associated with the membrane (*Figure 2—figure supplement 1*), suggesting that its inability to undergo cleavage was not because of defective membrane localisation. These results show that the MRE β-loop plays a critical role in the proteolytic function of RseP and that RseP selectively cleaves type II membrane proteins, even those with a small periplasmic domain.

## Pro substitutions in the MRE β-loop lead to the differential cleavage of the model substrates

Cys-scanning mutagenesis was performed to identify functionally important residues in the MRE β-loop. We constructed single Cys RseP derivatives by substituting Cys residues in place of the 15 residues in the MRE β-loop of Cys-less RseP-HM, which contained Ala residues in place of the 2 original Cys residues (Cys-33 and Cys-427). Complementation and model substrate cleavage assays showed that all the single Cys mutants, except G68C supported normal cell growth and cleaved HA-MBP-RseA(LY1)148 (*Figure 2—figure supplement 2*). Although the G68C mutant was inactive in complementation, it cleaved the model substrate with a slightly lower but significant efficiency. These results indicate that no residue in the MRE β-loop is specifically required for the proteolytic function of RseP. It is unclear why the G68C mutant failed to complement the Δ*rseP* mutation despite its substantial protease activity against the model substrate. G68C mutation might destabilise or inactivate the mutant protein under conditions employed in the complementation assay or might impair the cleavage of certain substrates that affect cell viability (see below).

We then performed a Pro-scanning experiment to test whether the secondary or tertiary structure of the MRE β-loop affects RseP function. Introduction of a Pro residue would disturb the higher-order structures of the MRE β-loop because Pro lacks the amide proton unlike other amino acids. Because the MRE β-loop originally had 1 Pro residue at position 65, we replaced other residues with Pro individually. In contrast to the results for Cys mutants, Pro substitution at positions 67, 68, 69 and 70, which are thought to be located in the distal part (Leu-66 to Glu-75) of the MRE β-loop, abolished complementation (*Figure 2B*). Consistently, the cleavage of HA-MBP-RseA148 was impaired moderately (G67P) or severely (G68P, Y69P and V70P) by these mutations (*Figure 2C*). Similar cleavage defects were observed with HA-MBP-RseA(LY1)148 for G67P, Y69P and V70P mutants (*Figure 2D*). Interestingly, the G68P mutant, which was almost completely defective in cleaving HA-MBP-RseA148, efficiently cleaved HA-MBP-RseA(LY1)148.

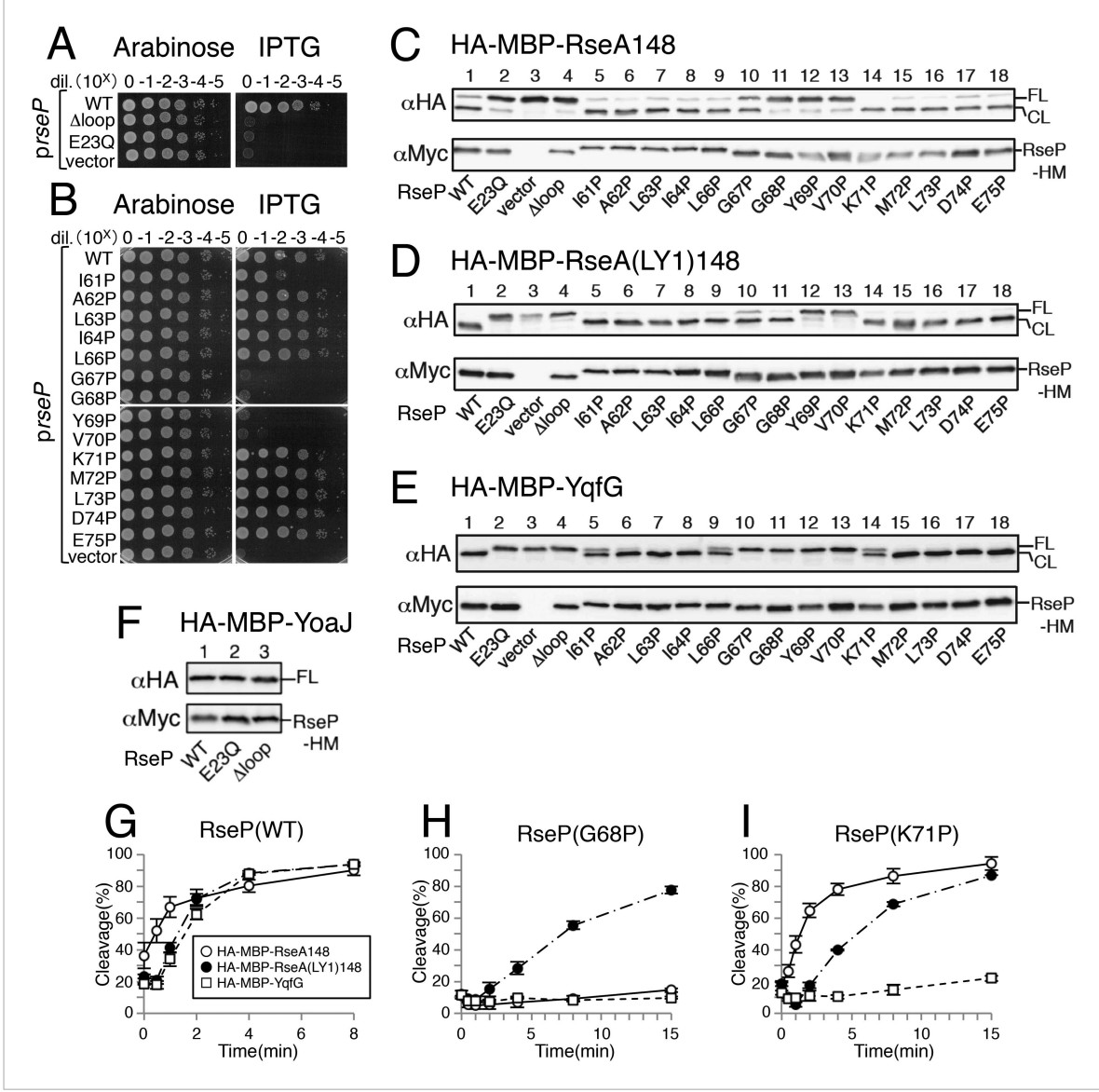

**Figure 2**. Effects of Pro substitutions in the MRE β-loop on RseP function. (**A** and **B**) Complementation assay. Strain KK31 [*rseP*::*kan*/pKK6 (P*ara-rseP*)] harbouring a plasmid encoding the indicated mutant form of RseP-His$_6$-Myc (RseP-HM) under the *lac* promoter or the corresponding vector was grown in L medium containing 0.02% arabinose. The cultures were serially diluted with saline, and 4 μl of the diluted cultures were spotted on L agar plates containing 0.02% arabinose or 1 mM IPTG. The plates were incubated at 30°C for 20 hr. (**C–F**) Immunoblotting analysis of substrate cleavage. KA306 (Δ*rseA*/Δ*rseP*/Δ*clpP*) cells harbouring a plasmid encoding the indicated model substrate was transformed with plasmids encoding the indicated mutant forms of RseP-HM and were grown at 30°C in M9-based medium containing 1 mM IPTG and 1 mM cAMP for 3 hr. Proteins were precipitated using TCA, solubilised in 1% SDS and analysed by 10% Laemmli–SDS-PAGE and immunoblotting with anti-HA or anti-Myc antibody. FL and CL indicate full-length and RseP-cleaved forms, respectively, of each model substrate. (**G–I**) Pulse-chase analysis of substrate cleavage. KA306 cells harbouring an appropriate combination of plasmids encoding the indicated RseP-HM mutant and the model substrate were grown in M9-based medium containing 1 mM IPTG at 30°C. The cells were labelled with [$^{35}$S]-methionine for 30 s and were chased with unlabelled methionine as indicated. Proteins were immunoprecipitated using agarose-conjugated anti-HA antibody and were analysed by 10% Laemmli–SDS-PAGE. Cleavage (%) was calculated using the following equation: cleavage (%) = 100 × (CL)/[(FL) + (CL)], where FL and CL are the intensities of the respective bands that were corrected for methionine content. Two independent experiments were performed, and mean values are shown along with standard deviations. See *Figure 2—source data 1* for gel images and quantitated band intensities data for *Figure 2G–I*. See *Figure 2—figure supplement 1* for integral association of HA-MBP-YqfG and HA-MBP-YoaJ with membrane. See *Figure 2—figure supplement 2* for complementation and model substrate cleavage activity of the RseP derivatives with a Cys substitution in the MRE β-loop.

The following source data and figure supplements are available for figure 2:

**Source data 1**. Zip file containing gel images and quantified band intensity data for the pulse-chase experiments.

*Figure 2. continued on next page*

*Figure 2. Continued*

**Figure supplement 1**. Membrane localization of the model substrates.

**Figure supplement 2**. Effects of the MRE β-loop Cys substitutions on the RseP function.

Pro mutations in the MRE β-loop exerted more profound effects on the cleavage of HA-MBP-YqfG than on the cleavage of the other 2 model substrates. Cleavage was severely impaired by G67P, G68P, Y69P and V70P mutations and was moderately impaired by I61P, L66P and K71P mutations. However, the latter 3 mutations did not inhibit the cleavage of the other 2 model substrates to detectable levels.

Substrate cleavage kinetics was investigated by pulse-chase experiments (*Figure 2G–I* and *Figure 2—source data 1*). We found that the initial rates of HA-MBP-RseA(LY1)148 and HA-MBP-YqfG cleavage by co-expressed RseP-HM were slightly lower than that of HA-MBP-RseA148 cleavage (*Figure 2G*). During the 15 min chase period, the G68P mutant significantly cleaved HA-MBP-RseA(LY1)148 (*Figure 2H*) while the K71P mutant efficiently cleaved HA-MBP-RseA148 and HA-MBP-RseA(LY1)148 but only slightly cleaved HA-MBP-YqfG, which was consistent with immunoblotting results (*Figure 2I*). Thus, the intrinsic susceptibility of each model substrate to wild-type RseP was not correlated with its susceptibility to MRE β-loop mutants, indicating that the MRE β-loop mutations resulted in the differential cleavage of different model substrates.

## The MRE β-loop is important for stable substrate–RseP interaction

We investigated the possible role of the MRE β-loop in RseP–substrate interaction by co-immunoprecipitation experiments (*Figure 3*). Inverted membrane vesicles (IMVs) were prepared from Δ*rseA*/Δ*rseP* strain expressing HA-MBP-RseA and RseP-HM derivatives, solubilised with *n*-dodecyl-β-D-maltoside (DDM) and were subjected to immunoprecipitation with agarose-conjugated anti-Myc or anti-HA antibody. The precipitated proteins were analysed by anti-HA or anti-Myc immunoblotting. In all the following co-immunoprecipitation/crosslinking experiments, RseP derivatives carried a mutation (E23Q) in the active site motif to prevent substrate cleavage during expression and immunoprecipitation/crosslinking. We previously used a similar approach to show that several residues in TM3, including Asn-389, play important roles in substrate binding (*Koide et al., 2008*). Consistent with the previous results, wild-type RseP-HM was co-immunoprecipitated with HA-MBP-RseA and HA-MBP-RseA was co-immunoprecipitated with RseP-HM, but no co-immunoprecipitation was observed with RseP-HM having an N389L mutation (*Figure 3A*). Deletion of the MRE β-loop almost completely abolished co-immunoprecipitation, suggesting its importance in RseP–substrate interaction (*Figure 3B*). We found that all the MRE β-loop Pro mutants, except D74P located in the C-terminus of the loop, significantly decreased co-immunoprecipitation efficiency, indicating that the Pro mutations compromised the interaction between RseP-HM and model substrates (*Figure 3A*). The result that absence of or mutations in the MRE β-loop interfered with RseP–substrate interaction suggests that the MRE β-loop is required for stable substrate binding by RseP. Pro substitutions in the proximal part (Ile-61 to Ile-64) of the MRE β-loop had little effect on substrate proteolysis (*Figure 2*) and decreased co-immunoprecipitation. These mutants would exhibit weak but sufficient interaction with substrates to promote cleavage in a membrane-integrated state.

We then examined the interaction of RseP with YoaJ (*Figure 3—figure supplement 1*). Anti-HA co-immunoprecipitation showed that RseP-HM was pulled down with HA-MBP-YoaJ in an MRE β-loop-dependent manner, indicating that YoaJ can interact with RseP although it is not cleaved by RseP and that the MRE β-loop can affect this interaction. However, overexpression of HA-MBP-YoaJ little affected the cleavage of co-expressed HA-RseA148 by RseP. It would be possible that YoaJ interacts with the MRE β-loop with much lower affinity than RseA does in the membrane, although similar amounts of RseP-HM were co-immunoprecipitated with HA-MBP-YoaJ and HA-MBP-RseA148 after membrane solubilisation. Alternatively, YoaJ might bind to RseP at a site different from that for RseA and other substrate proteins. In this case, deletion of the MRE β-loop might have indirectly disturbed the former site. The nature of the RseP–YoaJ interaction should be clarified in future study.

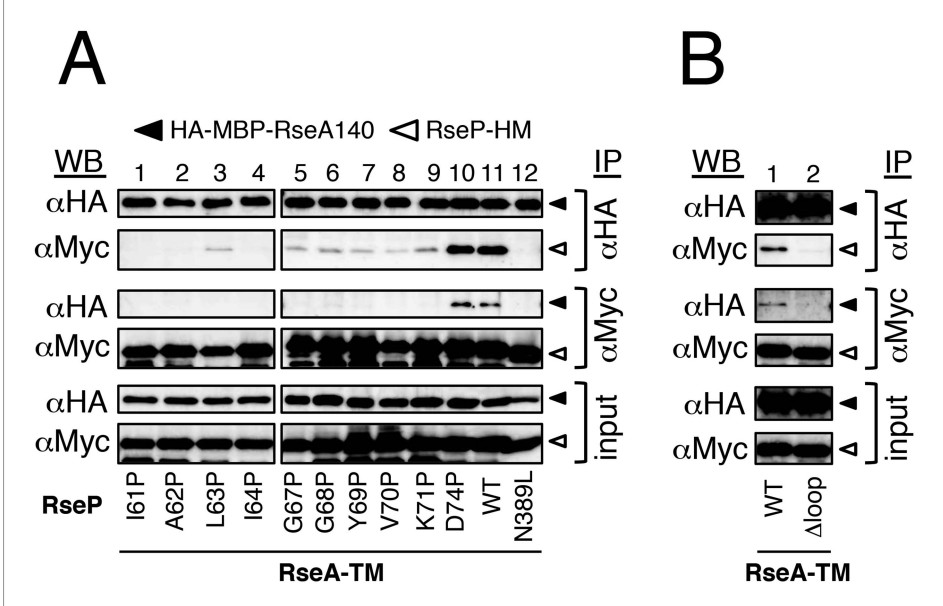

**Figure 3**. Effects of MRE β-loop mutations on RseP–substrate interaction. (**A** and **B**) Co-immunoprecipitation assays of RseP–substrate interaction. IMVs were prepared from KK211 (Δ*rseA*/Δ*rseP*) cells harbouring a plasmid encoding the indicated mutant of RseP-HM and pSTD881 (HA-MBP-RseA140). The IMVs were solubilised with DDM and were subjected to immunoprecipitation with anti-HA or anti-Myc antibody. The immunoprecipitates and the DDM-solubilised proteins (input) were analysed by 12.5% Laemmli–SDS-PAGE and immunoblotting with anti-HA and anti-Myc antibodies. Proteins form approximately fourfold more number of cells were loaded on the gel for the immunoprecipitated samples as compared with the input samples. All the RseP-HM derivatives carried the E23Q mutation. See *Figure 3—figure supplement 1* for interaction between RseP-HM and HA-MBP-YoaJ.

The following figure supplement is available for figure 3:

**Figure supplement 1**. Analysis of RseP–YoaJ interaction.

## In vivo photo-crosslinking between RseP and RseA

The above results and the presumed location of the MRE β-loop near the proteolytic active site suggest that the loop is directly involved in substrate binding by RseP. This was examined by site-directed in vivo photo-crosslinking experiments (*Figure 4*). Amber suppression-mediated incorporation of *p*-benzoylphenylalanine (*p*BPA), a non-natural photoreactive amino acid (*Young et al., 2010*), allowed the expression of RseP-HM derivatives with *p*BPA in the MRE β-loop. We examined the crosslinking of plasmid-expressed RseP-HM with chromosomally encoded RseA. We used Δ*rseP*/Δ*ompA*/Δ*ompC* strain as the host strain because *rseP* can be deleted in *rseA*[+] background when genes encoding outer membrane proteins OmpA and OmpC are deleted (*Douchin et al., 2006*). The host strain expressed DegS, which cleaved some amount of RseA to RseA148. Expression of *p*BPA-containing RseP-HM proteins decreased the accumulation of RseA148, indicating the functionality of these proteins, although some exhibited lower proteolytic activity against RseA (*Figure 4—figure supplement 1A*). After the exposure of cells to UV irradiation, the proteins were analysed by SDS-PAGE and immunoblotting. Anti-RseA immunoblotting showed UV irradiation produced bands of approximately 71 kDa (XL) when *p*BPA was incorporated at position 69, 71 or 74, suggesting that these bands represented photo-crosslinked products between RseP-HM and RseA (*Figure 4A*, upper panels). However, these bands were not detected by anti-Myc immunoblotting (*Figure 4A*, lower panels). As the expression level of plasmid-expressed RseP-HM was much higher than that of chromosomally encoded RseA, it seemed likely that only a small portion of RseP-HM was crosslinked with RseA. The over-expression of RseP-HM caused an increased background, making it difficult to detect weak bands of crosslinked products by anti-Myc immunoblotting. To confirm whether the 71 kDa band represented an RseP–RseA crosslinked product, we conducted immunoprecipitation

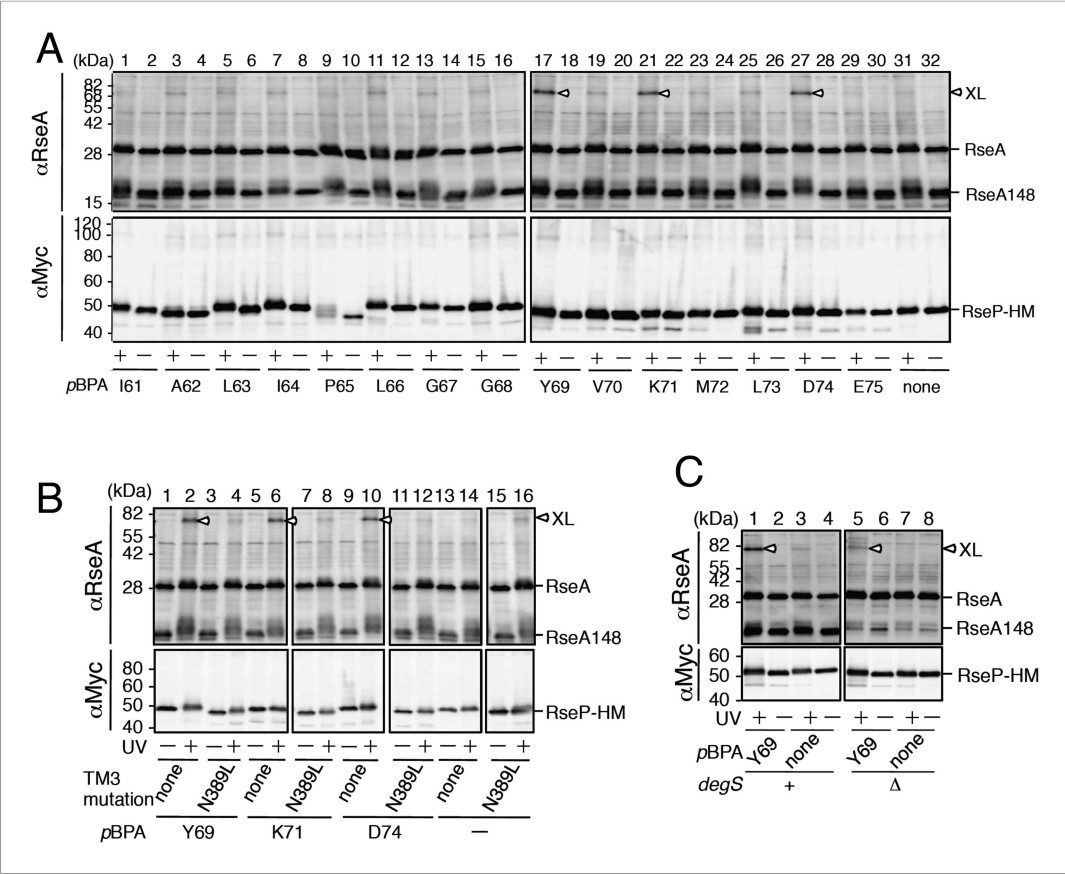

**Figure 4**. In vivo photo-crosslinking between RseP and RseA. (**A**) Analysis of the MRE β-loop/RseA crosslinking. KA418 (ΔompA/ΔompC/ΔrseP)/pEVOL-pBpF cells were transformed with a plasmid encoding an RseP derivative with an amber mutation at the indicated position. The cells were grown at 30°C in M9-based medium containing 0.02% arabinose and 1 mM pBPA for 6 hr and were irradiated with UV light for 0 or 10 min. Proteins were precipitated using TCA and were analysed by SDS-PAGE with a 10% wide-range gel (Nacalai Tesque, Inc. Kyoto, Japan) containing 1% SDS and immunoblotting with anti-RseA antibody or by SDS-PAGE with a 10% Laemmli gel and immunoblotting with anti-Myc antibody. XL indicates crosslinked products. (**B**) Effect of the N389L mutation on photo-crosslinking. KA418/pEVOL-pBpF strain was transformed with a plasmid encoding an RseP derivative with an amber mutation at the indicated position with or without the N389L mutation. The cells were subjected to photo-crosslinking analysis, as indicated in **A**. (**C**) Effect of periplasmic truncation of RseA on photo-crosslinking. KA418/pEVOL-pBpF or KA438 (KA418, ΔdegS)/pEVOL-pBpF cells were transformed with a plasmid encoding an RseP derivative with an amber mutation at the indicated position. The cells were subjected to photo-crosslinking analysis, as indicated in **A**. All the RseP-HM derivatives in **A–C** carried the E23Q mutation. See *Figure 4—figure supplement 1* for proteolytic activity of the pBPA-containing RseP derivatives and verification of RseP–RseA photo-crosslinking.

The following figure supplement is available for figure 4:

**Figure supplement 1**. Functionality of the pBPA-containing RseP derivatives and characterization of the photo-crosslinked product.

experiments by using total proteins obtained from UV-irradiated RseP(Y69pBPA)-HM-expressing cells (*Figure 4—figure supplement 1B*). Anti-RseA antibody precipitated a predominant 71 kDa band that reacted with both anti-RseA and anti-RseP antibodies (upper panels). Anti-Myc antibody also precipitated a single anti-RseA-reactive protein of 71 kDa (lower left panel). Although no clear band of the corresponding size was obtained after anti-RseP immunoblotting (lower right panel), this may be due to increased background disturbance because of the high expression of RseP-HM. These results further support the notion that the 71 kDa band was the RseP–RseA crosslinked product.

The N389L mutation that destabilised the RseP–substrate interaction considerably decreased RseA crosslinking at the 3 positions (*Figure 4B*), suggesting that the photo-crosslinking reflected the functional interaction between RseP and RseA. We examined the effect of the cleavage by DegS (*Figure 4C*). Periplasmic truncation of RseA was suppressed in the absence of DegS, thus increasing the accumulation of intact RseA, although few non-specific cleavages by other cellular proteases still occurred under this condition. RseP(Y69*p*BPA)-HM generated a significantly lower amount of the crosslinked product in the Δ*degS* strain than in the *degS⁺* strain, indicating that the MRE β-loop mainly interacts with the DegS-cleaved form of RseA.

## Disulfide crosslinking of the MRE β-loop with RseA

To determine whether the MRE β-loop directly interact with RseA TM, we examined disulfide crosslinking between RseP-HM and HA-RseA140 mimicking the DegS-cleaved form of RseA (*Kanehara et al., 2002*). An RseP-HM variant having a unique Cys residue at position 69, 70 or 71 was co-expressed with HA-RseA140, which has a unique Cys at position 109, or with its variant HA-RseA(A108C/C109A) 140, which has a unique Cys in place of Ala-108 (Cys-109 was replaced with Ala), in a Δ*rseA*/Δ*rseP*/ Δ*degS* strain. Ala-108 and Cys-109 are located in the middle of the RseA TM, and RseP-catalysed cleavage occurs between these residues (*Akiyama et al., 2004*; *Flynn et al., 2004*). Cells expressing these proteins were treated with $Cu^{2+}$(phenanthroline)$_3$ to induce disulfide bond formation. After quenching the oxidant, proteins were acid-denatured and solubilised in SDS. The samples were analysed directly by SDS-PAGE or after 2-mercaptoethanol treatment to cleave the disulfide bonds. The oxidant treatment generated a ∼75 kDa band that reacted with both anti-HA and anti-Myc antibodies when Y69C and K71C variants but not Cys-less and V70C variants of RseP-HM were co-expressed with HA-RseA(A108C/C109A)140 (*Figure 5A*). This band disappeared when the samples were treated with 2-mercaptoethanol before SDS-PAGE (*Figure 5B*). The same results were obtained when the single Cys RseP-HM variants were co-expressed with HA-RseA140(C109). These results were consistent with those of photo-crosslinking experiments because *p*BPA at positions 69 and 71 but not at position 70 was crosslinked with RseA, indicating that the MRE β-loop was near the RseP cleavage site. Together, our results strongly suggest that the MRE β-loop can directly bind to substrate TMs.

## Suppression of MRE β-loop mutations by destabilising substrate TM helices

We previously showed that helix-destabilising residues in substrate TMs promote the binding and cleavage of a substrate by RseP (*Akiyama et al., 2004*; *Koide et al., 2008*). Our current results showed

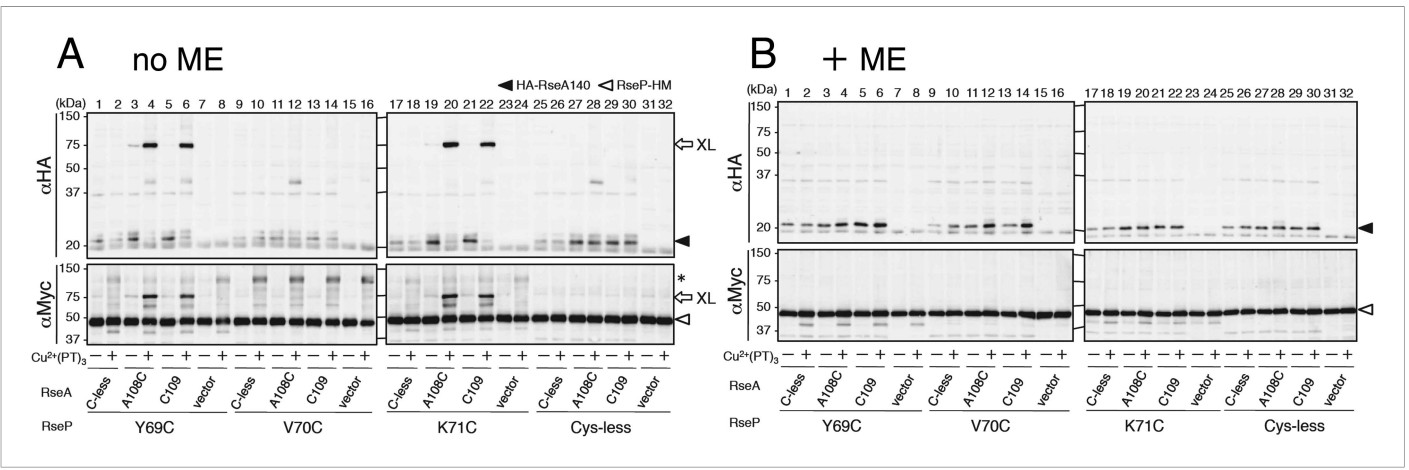

**Figure 5**. Disulfide crosslinking of RseP and RseA. AD1840 (Δ*rseA*/Δ*rseP*/Δ*degS*) cells harbouring a combination of plasmids encoding Cys-less or the indicated single Cys derivative of RseP-HM and HA-RseA140 were treated with $Cu^{2+}$(phenanthroline (PT))$_3$ at 37°C for 5 min. After quenching the oxidant, proteins were precipitated using TCA and free thiol groups were blocked by treatment with *N*-ethylmaleimide (NEM). The samples were analysed by 10% Laemmli–SDS-PAGE and immunoblotting with anti-HA (upper panels) or anti-Myc (lower panels) antibody, followed by treatment without (**A**) or with (**B**) 10% 2-mercaptoethanol (ME). All the RseP-HM derivatives carried the E23Q mutation. Asterisk indicates a possible dimer of RseP-HM (*Koide et al., 2008*) that was not characterised further.

that several Pro substitutions in the MRE β-loop exerted opposite effects, that is, they compromised substrate binding and cleavage. Therefore, we examined whether destabilisation of a substrate TM helix improves its cleavage by RseP MRE β-loop mutants. Because the MRE β-loop was inserted halfway into the membrane from the cytoplasmic side, we replaced each residue in the N-terminal region (Ile-8 to Phe-15) of the presumed YqfG TM by a Pro residue, a strong helix destabiliser, and investigated its cleavage by RseP G67P and K71P mutants, which showed severe and moderate defects, respectively, in cleaving HA-MBP-YqfG (*Figure 6*). Wild-type RseP-HM cleaved all the HA-MBP-YqfG mutants almost as efficiently as it cleaved the original HA-MBP-YqfG (*Figure 6A*). L11P and L12P mutations but not other mutations appreciably increased the cleavage by RseP(G67P)-HM (*Figure 6B*). In contrast, cleavage by RseP(K71P)-HM was markedly improved by mutations other than L12P and L13P. L12P and L13P also improved cleavage, but their effects were much lower (*Figure 6C*). We focused on 2 YqfG TM mutations I8P and L12P that resulted in differential cleavage by RseP(G67P)-HM and RseP(K71P)-HM and confirmed their effects by pulse-chase experiments (*Figure 6D–F* and *Figure 6—source data 1*). RseP(WT)-HM efficiently cleaved wild-type HA-MBP-YqfG and its I8P and L12P mutants (*Figure 6D*). G67P mutation in the RseP MRE β-loop severely impaired its ability to cleave wild-type HA-MBP-YqfG. Only some cleavage was observed after the 60 min chase period (*Figure 6E*). K71P mutation, which had a weaker effect on immunoblotting, also impaired the ability of RseP to cleave wild-type HA-MBP-YqfG during the 15 min chase period (*Figure 6F*). RseP(G67P)-HM significantly cleaved HA-MBP-YqfG (L12P) but not HA-MBP-YqfG(I8P) (*Figure 6E*) and RseP(K71P)-HM efficiently cleaved HA-MBP-YqfG (I8P) but not HA-MBP-YqfG(L12P) (*Figure 6F*), which was consistent with the immunoblotting results. Replacement of Leu-12 in YqfG by Asn, another strong helix-destabilising amino acid, also promoted cleavage by the RseP MRE β-loop Pro mutants. However, replacement of the same residue by a helix-forming Trp residue resulted in no increase in cleavage (*Figure 6A–C*), suggesting that helix destabilisation is important for improving substrate cleavage. The I8P or L12P mutation did not promote cleavage by the RseP Δloop variant, indicating that these mutations did not bypass the RseP MRE β-loop function for substrate cleavage (*Figure 6—figure supplement 1*).

Similar effects of a Pro substitution was also observed for LY1, another substrate TM (*Figure 6G*); the F21P mutation in LY1 significantly improved the cleavage of this TM by RseP(V70P)-HM that was almost inactive in cleavage of wild type LY1, but the Pro substitution of the neighbouring residue, Phe-20, did not.

The results that helix-destabilising mutations in substrate TMs at least partially suppress the defects in substrate cleavage induced by Pro substitutions in the RseP MRE β-loop in an allele-specific manner suggests a specific interaction between substrate TMs and the RseP MRE β-loop.

## Discussion

RseP is one of the most extensively characterised proteins among the S2P family of I-CLiPs. However, detailed mechanisms underlying its substrate recognition and cleavage are not completely understood. We focused on the MRE β-loop, a conserved intramembrane β-hairpin-like structure, and investigated its role in substrate proteolysis by RseP. Deletion or Pro substitution in the MRE β-loop impaired substrate cleavage, indicating its critical role in the proteolytic function of RseP. Crosslinking and co-immunoprecipitation experiments showed that the MRE β-loop was directly involved in substrate binding. Several Pro substitutions differentially affected RseP-catalysed proteolysis of substrates. Helix-destabilising mutations in substrate TMs suppressed the defects caused by mutations in the MRE β-loop in an allele-specific manner. These results collectively suggest that the MRE β-loop specifically recognises substrate TMs.

The proximal part of the MRE β-loop has a short β-strand (Ile[72]-Leu-Leu) in the reported crystal structure of mjS2P (*Feng et al., 2007*). However, no secondary structure is assigned for the distal part of the MRE β-loop. Analysis of the secondary structure with STRIDE (*Frishman and Argos, 1995*) suggested that Val[80]-Ala-Met in the distal part forms a β-strand (*Figure 1B–D*). This region is in an extended conformation with the side chains facing the opposite sides in an alternate manner (as expected for a β-strand) suggesting that this region forms a β-strand. Sequence-based prediction suggested that the MRE β-loops of RseP and SpoIVFB also have β-strands in the corresponding regions (*Figure 1D*), suggesting that the MRE β-loop commonly assumes a β-hairpin-like structure.

Cys-scanning mutagenesis suggest that no specific residue in the MRE β-loop is essential for the proteolytic function of RseP because the Cys substitutions had little or only slight effect on the substrate cleavage. In contrast, substrate cleavage was severely impaired by Pro substitutions. These

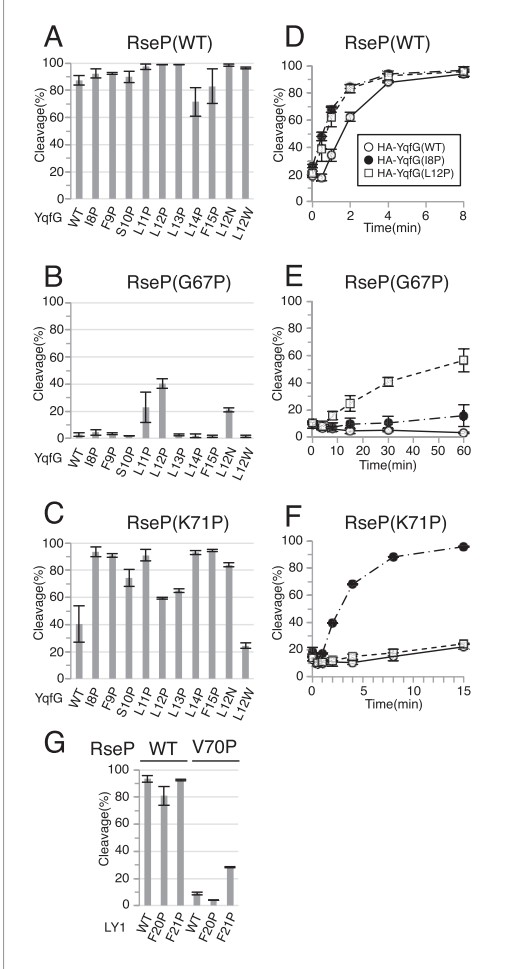

**Figure 6**. Suppression of the effect of RseP MRE β-loop mutations by destabilising substrate TM helix. (**A–C**) Immunoblotting analysis of YqfG cleavage. KA306 cells carrying a combination of plasmids encoding the indicated RseP-HM mutants and HA-MBP-YqfG mutants were grown and were subjected to immunoblotting analysis, as described in *Figure 2*. (**D–F**) Pulse-chase analysis of YqfG cleavage. KA306 cells harbouring a combination of plasmids encoding the indicated RseP-HM mutants and HA-MBP-YqfG mutants were grown and subjected to pulse-chase analysis, as described in *Figure 2*. Two independent experiments were performed, and mean values are shown along with standard deviations. See *Figure 6—source data 1* for gel images and quantitated band intensities data for *Figure 6D–F*. (**G**) Immunoblotting analysis of LY1 cleavage. KA306 cells carrying a combination of plasmids encoding the indicated RseP-HM mutants and HA-MBP-RseA(LY1)148 mutants were grown and were subjected to immunoblotting analysis, as above.

The following source data and figure supplement are available for figure 6:

**Source data 1**. Zip file containing gel images and quantified band intensity data for the pulse-chase experiments.

*Figure 6. continued on next page*

observations suggest that a higher-order structure of the MRE β-loop is important for the proteolytic function of RseP. Interestingly, some Pro substitutions in the MRE β-loop resulted in the differential cleavage of the 3 model substrates. Pulse-chase experiments showed that RseP(G68P)-HM significantly cleaved LY1 but only slightly cleaved RseA TM and YqfG TM. In contrast, RseP(K71P)-HM efficiently cleaved RseA TM and LY1 but only slightly cleaved YqfG TM. These 3 substrates were rapidly cleaved by wild-type RseP-HM. Therefore, the effects of MRE β-loop mutations cannot be simply explained by the intrinsic susceptibility of these substrates to RseP. In addition, some mutations in YqfG TM suppressed G67P and K71P mutations of the RseP MRE β-loop in an allele-specific manner. The I8P mutation improved the proteolysis by RseP(K71P)-HM but not by RseP(G67P)-HM, whereas the L12P mutation improved the proteolysis by RseP(G67P)-HM compared with that of RseP(K71P)-HM. Similarly the F21P, but not F20P, mutation in the LacY TM1 region (LY1) partially suppressed the V70P mutation of the RseP MRE β-loop. These results suggest that the MRE β-loop specifically recognises substrate TMs.

Direct evidence of MRE β-loop–substrate interaction was obtained by co-immunoprecipitation and crosslinking experiments. Pro substitutions in the MRE β-loop markedly decreased the co-immunoprecipitation of RseP-HM with HA-MBP-RseA after membrane solubilisation, suggesting that the integrity of the MRE β-loop is required for stable RseP–substrate interaction. Disulfide- and photo-crosslinking experiments showed that residues at positions 69 and 71 of the MRE β-loop contact with substrates. Cys residues at these positions disulfide-bonded with Cys residues on either side of the scissile bond in RseA. A similar result was reported recently for SpoIVFB and its substrate Pro-σ^K (*Zhang et al., 2013*). A Cys residue in place of Val-70 in the MRE β-loop of SpoIVFB formed disulfide bonds with Cys residues at several positions around the cleavage site in Pro-σ^K, suggesting that the MRE β-loop of SpoIVFB directly interacts with Pro-σ^K. However, the role of the MRE β-loop in SpoIVFB function has not been investigated experimentally. Although RseP also has a Val residue at the position corresponding to Val-70 in SpoIVFB (*Figure 1D*), we did not observe any substrate crosslinking at this position. This might be due to differences in the mode of enzyme–substrate interaction between RseP and SpoIVFB. Indeed, the substrates of these enzymes show some differences. In contrast to RseA, which is a single-spanning

*Figure 6. Continued*

**Figure supplement 1**. Cleavage of the I8P and L12P mutants of HA-MBP-YqfG depends on the MRE β-loop.

membrane protein with type II orientation, Pro-$\sigma^K$ is suggested to be peripherally associated with the membrane (*Zhou et al., 2013*). In addition, although helix-destabilising residues in substrate TMs are important for their efficient cleavage by RseP, these residues are not required for the cleavage of Pro-$\sigma^K$ by SpoIVFB (*Zhou et al., 2013*). Thus, the role of the MRE β-loop in substrate binding and cleavage might slightly differ between RseP and SpoIVFB.

The N389L mutation in RseP TM3, which was previously shown to weaken the RseP-substrate interaction (*Koide et al., 2008*), inhibited the in vivo photo-crosslinking of the MRE β-loop with RseA, suggesting that the observed crosslinking reflected a functional RseP–substrate interaction. The TM4 (corresponding to RseP TM3) of mjS2P that contains one of the active site residues (corresponding to Asp-402 in RseP TM3) can be located in the vicinity of the proteolytic active site and the MRE β-loop (*Feng et al., 2007*). We previously showed that Cys introduced at the positions of Pro-397 and Pro-399 in RseP TM3 can form a disulfide bond with Cys at multiple positions of RseA TM. In mjS2P, the residues corresponding to Pro-397 and Pro-399 of RseP reside in a loop-like structure. The region containing residues 397 and 399 might be flexible and act with the MRE β-loop in stable binding of a substrate.

An α-helix is generally not amenable to proteolysis and unwinds to an extended structure to undergo cleavage (*Wolfe, 2009*). Many zinc metalloproteinases have a β-strand (edge strand) close to their proteolytic active site that binds to substrates in an extended conformation through β-strand addition and presents them to catalytic residues (*Stocker and Bode, 1995*; *Langklotz et al., 2012*). Our results suggest that the MRE β-loop stabilises the extended conformation of a substrate by directly binding to it through β-strand addition, thus promoting substrate cleavage in a manner similar to the edge strand (*Figure 7*). Crosslinking at alternate positions in the Tyr$^{69}$-Val-Lys region supports the idea that this region is in a β-strand conformation during its interaction with a substrate. According to this model, mutations that disrupt the secondary structure of the distal part of the MRE β-loop would compromise its interaction with a substrate. Introduction of a Pro residue will disturb substrate accommodation through β-strand addition or interstrand interaction because Pro residues cannot serve as hydrogen bond donors because of the absence of the amide proton. Moreover, if the compromised interaction of the MRE β-loop with a substrate prevents efficient conformational conversion of the substrate TM and decreases its cleavage efficiency, destabilisation of the α-helical structure of the substrate TM might improve its cleavage by RseP MRE β-loop mutants, which is consistent with our results. A recent structural study of *E. coli* GlpG that belongs to the rhomboid family of I-CLiPs revealed that the P1–P4 region of a substrate forms a β-sheet with the GlpG loop 3 located near the active site (*Zoll et al., 2014*). Mutations in loop 3 severely compromise the GlpG activity, suggesting the interaction between the substrate and loop 3 is functionally important (*Baker and Urban, 2012*). Although the loop 3 of GlpG and the MRE β-loop of RseP differ in that the former is located near the periplasmic surface of the membrane and forms a parallel β-sheet with a substrate whereas the latter is located near the cytoplasmic surface and forms a anti-parallel β-sheet, they might have a similar role in stabilising an extended substrate structure and promoting its cleavage.

Differential effects of Pro substitutions in the MRE β-loop on the cleavage of the model substrates and allele-specific suppression of these mutations by helix-destabilising mutations in substrate TMs suggest that the MRE β-loop–substrate interaction occurs with some specificity. MRE β-loop-assisted cleavage of a substrate may be affected by helix stability and amino acid sequence of substrate TMs. We recently suggested that the periplasmic PDZ tandem of RseP acts as a size-exclusion filter to prevent the cleavage of substrates with bulky periplasmic domains (*Figure 7A*) (*Hizukuri et al., 2014*). In this study, we observed that RseP did not cleave YoaJ, a type II membrane protein with a very small periplasmic domain, suggesting the presence of additional mechanism(s) for substrate discrimination. In vivo photo-crosslinking experiments suggest that the MRE β-loop interacts with RseA after site 1 cleavage (*Figure 7B*) and that it acts as the second checkpoint for membrane proteins that passed the first check by the PDZ filter and contributes to their selective proteolysis. It should be noted that mjS2P does not have PDZ domains. Thus, the MRE β-loop could play a central role in cleaving specific substrates in this enzyme.

Although the MRE β-loop may not interact with intact RseA, our previous chemical crosslinking and co-immunoprecipitation experiments showed that RseP directly interacted with intact RseA in the membrane (*Kanehara et al., 2003*), suggesting that RseP has a separate binding site (exosite) for intact RseA. Because the PDZ filter excludes intact RseA from the intramolecular active site, the

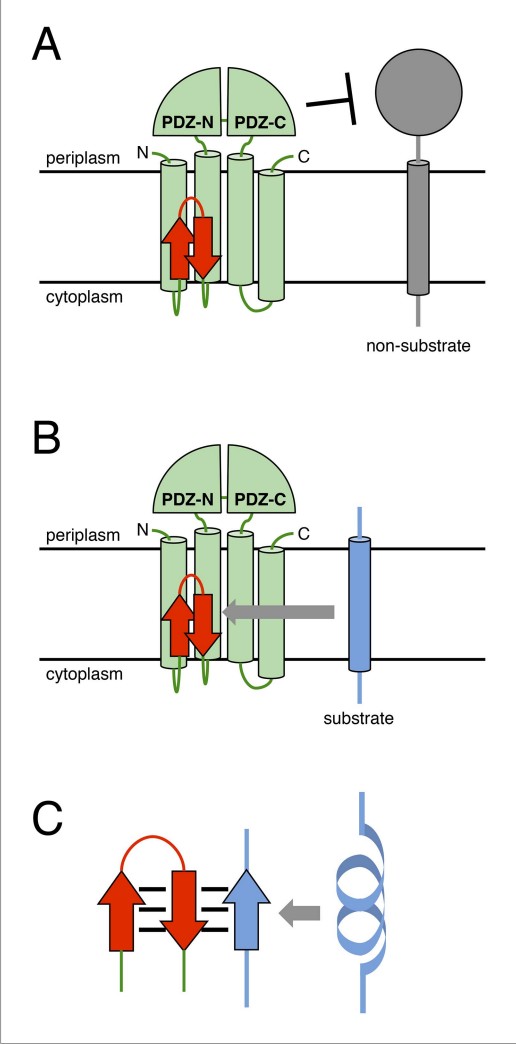

**Figure 7**. A model describing the function of the MRE β-loop. (**A**) The periplasmic PDZ tandem of RseP acts as a size-exclusion filter to discriminate between substrates based on the size of their periplasmic region. (**B**) The binding of the MRE β-loop serves as an additional checkpoint for substrates with a small periplasmic region and contributes to selective substrate cleavage. (**C**) The MRE β-loop binds to substrate TMs with some specificity because of β-strand addition, stabilises its extended conformation and presents the substrate to the recessed proteolytic active site of RseP.

presumed exosite may exist on the external surface of the enzyme. Other families of I-CLiPs such as rhomboid and γ-secretase have also been suggested to have an exosite (*Kornilova et al., 2005*; *Strisovsky et al., 2009*; *Watanabe et al., 2010*; *Arutyunova et al., 2014*). A systematic in vivo crosslinking approach would help in identifying binding sites for intact RseA. This along with the structural analysis of RseP, especially in complex with a substrate, would be essential for verifying our model and for understanding molecular mechanisms underlying substrate recognition and intramembrane proteolysis by RseP. Further, we would like to study the physiological significance of RseP-catalysed cleavage of YqfG and identify additional substrates of RseP to elucidate all the cellular roles of RseP.

## Materials and methods

### Bacterial strains

All the bacterial strains used in this study are derivatives of *E. coli* K12 and are listed in *Supplementary file 1*. KA306 was constructed by transferring the *clpP::cat* marker derived from MC4100 *clpP::cat* (provided by M. Kitagawa) into AD2328. The *clpP::cat* marker was introduced to stabilise the RseP-cleaved fragment of HA-MBP-RseA148. KA418 was constructed as follows. First, Δ*ompA::kan* from JW0940 (*Baba et al., 2006*) was introduced into CU141 by P1 transduction, and the *kan* cassette of the resulting strain was deleted using pCP20, as described previously (*Datsenko and Wanner, 2000*). Introduction of Δ*ompC::kan* from JW2203 (*Baba et al., 2006*) and deletion of the *kan* cassette were performed in a similar manner to yield YH426. KA363 was constructed by introducing *rseP::kan* from KK211 into YH426. We found that KA363 had spontaneously lost F'*lac*+ *lacI*q. We thus constructed KA418 by re-introducing F'*lac*+ *lacI*q into KA363 by conjugation. KA438 was constructed by transferring *degS::tet* from AD1839 (*Kanehara et al., 2002*) to KA418 by P1 transduction.

### Plasmids

Plasmids used in this study are listed in *Supplementary file 2*. pKA1 and pKA117 were constructed by replacing the region encoding the MRE β-loop (Ile[61] to Glu[75]) in *rseP-his_6-myc* on pKK49 or a corresponding region in *rseP(Cys-less)-his_6-myc* on pSTD892 with GGCGGT (encoding Gly–Gly) by site-directed mutagenesis. pKA19 was constructed by introducing the E23Q mutation into pSTD892. pKA52 was constructed by replacing a 1.5 kb BamHI–SacI fragment of pKA1 with a corresponding fragment of pKK34. pKA65 was constructed by cloning a 1.4 kb EcoRI–HindIII fragment of pYH19 in the same site of pSTD689. Other plasmids were constructed by site-directed mutagenesis by using appropriate combinations of primers and template plasmids. Mutations were confirmed by DNA

sequencing. pMZ14 was constructed by cloning a PCR-amplified fragment encoding YqfG into the SalI–PstI site of pSTD835 and by ligating an EcoRI–HindIII fragment of the resulting plasmid with EcoRI-/HindIII-digested pSTD689. pKA195 was constructed by cloning a 0.6 kb EcoRI–HindIII fragment of pYH18 in the same site of pSTD689. For construction of pKA210, first, a PCR-amplified fragment encoding YoaJ was cloned in the SalI–PstI site of pSTD835, and then an EcoRI–HindIII fragment of the resulting plasmid was ligated with EcoRI-/HindIII-digested pSTD689. pKA268 was constructed by cloning a 1.4 kb EcoRI–HindIII fragment of pMZ14 in the same site of pUC118.

## Media

L medium (10 g/l bactotryptone, 5 g/l yeast extract and 5 g/l NaCl; pH adjusted to 7.2 by using NaOH) and M9 medium (without $CaCl_2$) (*Miller, 1972*) were used for bacterial cultivation. Ampicillin (50 µg/ml), chloramphenicol (20 µg/ml), spectinomycin (50 µg/ml), kanamycin (25 or 12.5 µg/ml) and/or tetracycline (25 µg/ml) were added for selecting transformants and transductants and for growing plasmid-harbouring strains.

## Cell fractionation and alkali extraction

Cells were grown at 30°C in L medium containing 1 mM isopropyl-β-D-thiogalactopyranoside (IPTG) and cAMP and were suspended in 50 mM HEPES-KOH (pH 8.0) containing 50 mM KCl, 10% glycerol and 1 mM dithiothreitol (DTT). The cells were then fractionated after sonication or subjected to alkali extraction. For cell fractionation (*Kihara et al., 2001*), the cells were mixed with 1/10 vol of 1 mg/ml lysozyme in 100 mM EDTA (pH 8.0) and were incubated at 0°C for 30 min. The cells were then disrupted by sonication and ultracentrifuged (99,000×g for 1 hr at 4°C). The pellet was suspended in 50 mM HEPES-KOH (pH 8.0) containing 50 mM KCl, 10% glycerol and 1 mM DTT. Proteins were precipitated using 5% (final concentration) trichloroacetic acid (TCA). Alkali fractionation was performed as described previously (*Ito and Akiyama, 1991*). The cells were suspended in 50 mM HEPES-KOH (pH 8.0) containing 50 mM KCl, 10% glycerol and 1 mM DTT. After addition of 1/10 vol of 10 mg/ml lysozyme in 50 mM EDTA (pH 8.0), the cells were incubated at 0°C for 5 min and were disrupted by freezing-thawing. The samples were then mixed with an equal volume of cold 0.2 N NaOH, vortexed vigorously for approximately 10 s and ultracentrifuged in a microfuge (99,000×g for 1 hr at 4°C). The supernatant was mixed with 1/10 vol of 100% (wt/vol) TCA. Acid denatured protein precipitates were collected by centrifugation, washed with acetone, and analysed by 12.5% SDS-PAGE and immunoblotting. Because the chromosomally encoded HflD was expressed at a very low level and was difficult to detect, we expressed HflD from a multicopy plasmid.

## In vivo protease activity of RseP

Cells harbouring an appropriate combination of plasmids encoding an RseP derivative and a model substrate were grown at 30°C in M9 medium supplemented with 20 µg/ml of each of the 20 amino acids, 2 µg/ml thiamine, 0.4% glucose, 1 mM IPTG and 1 mM cAMP for 3 hr. A part of the culture was mixed with an equal volume of 10% TCA (*Hizukuri and Akiyama, 2012*). Protein precipitates were recovered by centrifugation, washed with acetone and dissolved in 1× SDS sample buffer. Proteins were analysed by 10% Laemmli–SDS-PAGE and immunoblotting with anti-Myc and anti-HA antibodies, as described previously (*Inaba et al., 2008*). The blotted proteins were visualised and quantified using ECL or ECL Prime Western Blotting Detection Reagent (GE Healthcare, Waukesha, WI) and LAS-3000 Mini Lumino-Image Analyzer (Fujifilm, Tokyo, Japan).

## Pulse-chase experiments

Cells harbouring an appropriate combination of plasmids encoding an RseP derivative and a model substrate were grown at 30°C in M9 medium supplemented with 20 µg/ml of each of 18 amino acids (except Met and Cys), 2 µg/ml thiamine and 0.4% glucose to a mid-log phase and were induced with 1 mM IPTG and 5 mM cAMP for 10 min. The cells were then labelled with 370 kBq/ml [$^{35}$S]-methionine (American Radiolabeled Chemicals, St. Louis, MO) for 30 s. Chasing was performed using 200 µg/ml unlabelled methionine for the indicated periods. Proteins were precipitated using 5% (final concentration) TCA, washed with acetone, dissolved in 50 mM Tris–HCl (pH 8.1) containing 1 mM EDTA and 1% SDS and immunoprecipitated with anti-HA antibody, as described previously (*Akiyama et al., 2004*). Labelled

proteins were separated by SDS-PAGE and were visualised and quantified using a phosphor imager (BAS1800) (Fujifilm).

## Co-immunoprecipitation assay

Co-immunoprecipitation experiments were performed as described previously (Koide et al., 2008). Total membranes were suspended in 50 mM HEPES-KOH (pH 7.5) containing 50 mM KCl and 20% glycerol, diluted 10-fold with 50 mM HEPES-KOH (pH 7.5) containing 300 mM KCl and 10% glycerol and solubilised with 1% DDM on ice for 1 hr. After clarification, the supernatant was incubated with agarose-conjugated mouse monoclonal anti-HA (F-7) or anti-Myc (9E10) antibody (Santa Cruz Biotechnology, Inc. Santa Cruz, CA) at 4°C for 3.5 hr with rotation. Immunocomplexes were collected, washed 3 times with 50 mM HEPES-KOH (pH 7.5) containing 300 mM KCl, 10% glycerol and 0.1% DDM and dissolved in 1× SDS sample buffer. The samples were analysed by 12.5% Laemmli–SDS-PAGE and by immunoblotting with rabbit polyclonal anti-Myc and anti-HA antibodies (Santa Cruz Biotechnology, Inc.).

## In vivo crosslinking experiments

Cells harbouring a plasmid encoding an RseP derivative with an amber mutation in the MRE β-loop region and pEVOL-pBpF (Addgene, Inc., Cambridge, MA) were grown at 30°C in M9 medium supplemented with 2 μg/ml thiamine, 0.4% glucose, 0.02% L-arabinose and 1 mM pBPA for 6 hr. After adding 100 μg/ml (final concentration) spectinomycin to halt protein synthesis, the cells were irradiated with UV light (365 nm) at 4°C for 10 min, as described previously (Narita et al., 2013). Proteins were precipitated by mixing the UV-irradiated cells with 1/20 vol of 100% TCA, washed with acetone, dissolved in 1× SDS sample buffer and analysed by 10% Laemmli–SDS-PAGE and immunoblotting with anti-RseA and anti-Myc antibodies.

For verifying the in vivo crosslinked products, TCA-precipitated proteins were dissolved in 50 mM Tris–HCl (pH 8.1) containing 1% SDS and 1 mM EDTA, diluted 33-fold with 50 mM Tris–HCl (pH 8.1) containing 150 mM NaCl and 1% NP-40 and immunoprecipitated with TrueBlot Anti-Rabbit IgG IP Beads (eBioscience, Inc., San Diego, CA) plus anti-RseA antibody or agarose-conjugated anti-Myc antibody (Akiyama et al., 2004). The proteins were separated by SDS-PAGE on a 10% wide-range gel (Nacalai Tesque, Inc., Japan) and by immunoblotting with anti-RseA and anti-RseP antibodies by using Can Get Signal Immunostain Enhancer Solution (Toyobo, Co., Ltd, Osaka, Japan) and TrueBlot Anti-Rabbit IgG.

## Disulfide crosslinking

Cells harbouring an appropriate combination of plasmids encoding an RseP derivative and a model substrate were grown at 30°C in L medium containing 1 mM IPTG and 1 mM cAMP for 3.5 hr. Disulfide bond formation was induced as described previously (Koide et al., 2008). Briefly, the cells were treated with 0.1 mM $Cu^{2+}$(phenanthroline)$_3$ or 3 mM 2-phenanthroline at 37°C for 5 min. Oxidation was terminated by incubating the cells with 12.5 mM neocuproine. Proteins were precipitated using TCA and dissolved in 100 mM Tris–HCl (pH 7.5) containing 1.5% SDS, 5 mM EDTA and 25 mM NEM. The samples were mixed with an equal volume of SDS sample buffer containing no or 10% 2-mercaptoethanol, boiled at 98°C for 5 min and analysed by 10% Laemmli–SDS-PAGE and immunoblotting with anti-Myc and anti-HA antibodies.

## Acknowledgements

We thank M Kitagawa and National BioResource Project E. coli for the E. coli strains. We also thank H Ebise and K Yoshikaie for providing technical assistance.

## Additional information

### Funding

| Funder | Grant reference | Author |
|---|---|---|
| Ministry of Education, Culture, Sports, Science, and Technology (MEXT) | Grant-in-Aid for Scientific Research on Innovative Areas 24117003 | Hiroyuki Mori |

| Funder | Grant reference | Author |
|---|---|---|
| Japan Society for the Promotion of Science (JSPS) | Grant-in-Aid for Scientific Research (B) 24370054 | Yoshinori Akiyama |
| Japan Society for the Promotion of Science (JSPS) | Grant-in-Aid for Scientific Research (B) 26291016 | Terukazu Nogi |
| Japan Society for the Promotion of Science (JSPS) | Grant-in-Aid for Scientific Research (B) 15H04350 | Yoshinori Akiyama |
| Ministry of Education, Culture, Sports, Science, and Technology (MEXT), and Japan Agency for Medical Research and Development (AMED) | Platform for Drug Discovery, Informatics, and Structural Life Science | Terukazu Nogi |
| Kyoto University | Joint Usage/Research Center program, Institute for Virus Research | Yoshinori Akiyama |
| Ministry of Education, Culture, Sports, Science, and Technology (MEXT) | Grant-in-Aid for Scientific Research on Innovative Areas 15H01532 | Yoshinori Akiyama |
| Japan Society for the Promotion of Science (JSPS) | Grant-in-Aid for Young Scientists (B) 26840033 | Yohei Hizukuri |

The funders had no role in study design, data collection and interpretation, or the decision to submit the work for publication.

### Author contributions

KA, Conception and design, Acquisition of data, Analysis and interpretation of data, Drafting or revising the article; SM, Conception and design, Acquisition of data, Analysis and interpretation of data; YH, HM, TN, Conception and design, Analysis and interpretation of data; YA, Conception and design, Analysis and interpretation of data, Drafting or revising the article

## Additional files

### Supplementary files

• Supplementary file 1. Table S1. Strains used in this study.

• Supplementary file 2. Table S2. Plasmids used in this study.

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
