## [Decision Letter]

Thank you for submitting your work entitled “Roles of the membrane-reentrant β-hairpin-like loop of RseP protease in selective substrate cleavage” for peer review at *eLife*. Your submission has been favorably evaluated by Vivek Malhotra (Senior Editor) and three reviewers, one of whom is a member of our Board of Reviewing Editors.

The reviewers have discussed the reviews with one another and the Reviewing editor has drafted this decision to help you prepare a revised submission.

The reviewers agree that this manuscript is broadly suitable for publication in *eLife*. It addresses a fundamental issue of high significance and broad interest. The issues that all agree need addressing before it could be accepted are as follows:

1) How general is the compensatory effect of the helix-destabilizing mutations in the substrate? Do they have a similar effect when introduced into YqfG or RseA? Testing the effect of helix destabilizing mutants would shed further light on the mechanism.

2) Does RseP interact with YoaJ, a non-substrate? If it does, can YoaJ inhibit cleavage of RseA or YqfG? In other words, is the binding site in the MRE β-loop separated from another binding site for the rest of the transmembrane domain, in a mechanistic analogy to what was proposed for rhomboids (Strisovsky et al., Mol Cell 2009)?

3) Comparisons with rhomboid mechanisms should be discussed in more detail. The first evidence for an exosite was published by Strisovsky et al., and a recent paper with a structure of a rhomboid in complex with a peptide substrate has suggested that the L3 loop might have a similar role.

4) In Figure 1 we are presented with a model of the MRE β−loop. The structure of this region is important since residues along on face of the strand 68-71 appear to be involved in substrate recognition, as depicted in Figure 7. According to the Methods, secondary structure was determined using STRIDE but the figure legend states this model is based on the crystal structure of mjS2P (Figure 1). In the manuscript, the Methods section needs to be adjusted to precisely reflect how this model was generated. What software was used to generate this model? Please clarify the section of the manuscript where the model is discussed.

5) Previous work in the lab has identified helix 3 of RseP as important in substrate recognition. This should be brought into the Discussion and placed in context of where TM3 resides in relation to the MRE β−loop.

---

## [Author Response]

1) How general is the compensatory effect of the helix-destabilizing mutations in the substrate? Do they have a similar effect when introduced into YqfG or RseA? Testing the effect of helix destabilizing mutants would shed further light on the mechanism.

To address the above problem, we first constructed HA-MBP-RseA148 derivatives with a Pro substitution in the RseA TM region, but we found that many of these mutants were not cleaved even by wild type RseP. As the RseA TM has unusually low hydrophobicity (for example, TMHMM program does not predict the RseA TM as a “true” TM segment), we suppose that the Pro substitutions would interfere with membrane integration. We thus used HA-MBP-RseA(LY1)148. We found that the F21P mutation in LY1 significantly improved the cleavage of LY1 by RseP(V70P)-HM, an MRE β-loop mutant almost defective in cleavage of wild type LY1, but the Pro substitution of the neighbouring residue Phe-20 did not, suggesting that Pro substitutions could generally suppress the MRE β-loop mutations in an allele-specific manner. These points were described in Results (subsection “Suppression of MRE β-loop mutations by destabilising substrate TM helices”) and Discussion (fourth paragraph) and the data was presented as Figure 6 in the revised manuscript.

2) Does RseP interact with YoaJ, a non-substrate? If it does, can YoaJ inhibit cleavage of RseA or YqfG? In other words, is the binding site in the MRE β-loop separated from another binding site for the rest of the transmembrane domain, in a mechanistic analogy to what was proposed for rhomboids (Strisovsky et al., Mol Cell 2009)?

Co-immunoprecipitation experiments showed that HA-MBP-YoaJ can interact with RseP. On the other hand, the HA-MBP-YoaJ did not stably interact with the RsePΔloop mutant, suggesting that the MRE β-loop might directly or indirectly affect the binding of YoaJ to RseP. However, overexpression of HA-MBP-YoaJ little affected the RseP-dependent cleavage of co-expressed RseA. It would be possible that YoaJ interacts with the MRE β-loop with much lower affinity than RseA does in the membrane, although similar amounts of RseP was pulled down with YoaJ and RseA after membrane solubilization. Alternatively, the binding sites for YoaJ and RseA (and other cleavable substrates) are not the same. In this case, the deletion of the MRE β-loop might indirectly affect the YoaJ-binding site. Further detailed investigation will be needed to characterize the nature of YoaJ-binding to RseP and we would like to address this issue in future study. These points were described in Results (subsection “The MRE β-loop is important for stable substrate–RseP interaction”) and the data was presented as Figure 3—figure supplement 1 in the revised manuscript.

*3) Comparisons with rhomboid mechanisms should be discussed in more detail. The first evidence for an exosite was published by Strisovsky et al., and a recent paper with a structure of a rhomboid in complex with a peptide substrate has suggested that the L3 loop might have a similar role*.

We thank the reviewers for the helpful suggestions. We now state that an exosite has been reported for Rhomboid and cite the study by Strisovsky et al. in the Discussion. We also discuss similarity of GlpG (*E. coli* rhomboid protease) L3 loop to the MRE β-loop in substrate binding/recognition.

*4) In*
Figure 1
*we are presented with a model of the MRE β−loop. The structure of this region is important since residues along on face of the strand 68-71 appear to be involved in substrate recognition, as depicted in*
Figure 7*. According to the Methods, secondary structure was determined using STRIDE but the figure legend states this model is based on the crystal structure of mjS2P (*Figure 1*). In the manuscript, the Methods section needs to be adjusted to precisely reflect how this model was generated. What software was used to generate this model? Please clarify the section of the manuscript where the model is discussed*.

We are sorry for the ambiguous description in the original manuscript. For Figure 1, the polypeptide backbone is simply extracted from the model in Figure 1 (the structure of mjS2P) and the amino acid residues of the corresponding region of RseP are assigned on the model. Secondary structure of the MRE β−loops (Figure 1) is assigned based on the analysis of amino acid sequences by PsiPred (for RseP and SpoIVFB) or the analysis of the crystal structure by STRIDE (for mjS2P). We clearly state these points in the legend to Figure 1, and in the Discussion.

*5) Previous work in the lab has identified helix 3 of RseP as important in substrate recognition. This should be brought into the Discussion and placed in context of where TM3 resides in relation to the MRE β−loop*.

The crystal structures of mjS2P have been reported. The TM4 (corresponding to RseP TM3) of mjS2P that contains one of the active site residues can be located in the vicinity of the proteolytic active site and the MRE β-loop. We have previously shown that Cys introduced at the positions of Pro-397 and Pro-399 in RseP TM3 can form a disulfide bond with Cys at multiple positions of RseA TM. In mjS2P, the residues corresponding to Pro-397 and Pro-399 of RseP reside in a loop-like structure in TM4. The region containing residues 397 and 399 might be flexible and act with the MRE β-loop in stable binding of a substrate. This point is described in the Discussion.